# QUALITY CONTROL AT YOUR FINGERTIPS: QUALITY-AWARE TRANSLATION MODELS

## ABSTRACT

Maximum-a-posteriori (MAP) decoding is the most widely used decoding strategy for neural machine translation (NMT) models. The underlying assumption is that model probability correlates well with human judgment, with better translations being more likely. However, research has shown that this assumption does not always hold, and decoding strategies which directly optimize a utility function, like Minimum Bayes Risk (MBR) or Quality-Aware decoding can significantly improve translation quality over standard MAP decoding. The main disadvantage of these methods is that they require an additional model to predict the utility, and additional steps during decoding, which makes the entire process computationally demanding. In this paper, we propose to make the NMT models themselves quality-aware by training them to estimate the quality of their own output. During decoding, we can use the model's own quality estimates to guide the generation process and produce the highest-quality translations possible. We demonstrate that the model can self-evaluate its own output during translation, eliminating the need for a separate quality estimation model. Moreover, we show that using this quality signal as a prompt during MAP decoding can significantly improve translation quality. When using the internal quality estimate to prune the hypothesis space during MBR decoding, we can not only further improve translation quality, but also reduce inference speed by two orders of magnitude.

## 1 INTRODUCTION

Most state-of-the-art models for Natural Language Processing (NLP) tasks are probabilistic, with the most frequent parameterization being based on neural networks. Once these models are trained, the prevailing decoding strategy for Natural Language Generation (NLG) is MAP decoding, i.e. select the hypothesis that maximizes the conditional probability given an input. As an exact maximization is computationally intractable due to the huge number of possible sentences, typically beam search or greedy decoding are used to approximate the search for the best hypothesis. Machine translation (MT) is a prominent example of these types of models, where the system is trained to generate a sentence in a target language given a source sentence in another language. Nonetheless, Eikema & Aziz (2020) have demonstrated that MAP decoding methods may be suboptimal due to the presence of misaligned probability distributions. Moreover, NMT models often assign human translations lower probabilities than their own beam search outputs due to calibration issues (Ott et al., 2018; Freitag et al., 2020).

Eikema & Aziz (2020; 2022) applied MBR decoding for NMT models as an alternative generation approach. MBR decoding follows a self-consistency approach by sampling from the model distribution and giving precedence to hypotheses that exhibit greater similarity to all other hypotheses. In contrast to MAP decoding, MBR decoding's objective is not centered on generating the translation with the highest estimated model probability. Instead, its focus lies in selecting the translation that exhibits the highest quality based on a utility metric. Subsequent research conducted by Freitag et al. (2022a) showed that MBR decoding with *neural* utility metrics leads to significant improvements over beam search decoding. However, MBR is computationally intensive, with a time complexity of $O(n^2)$ for a candidate list containing $n$ samples (ideally $n$=100 to $1\,000$ (Freitag et al., 2022a)), necessitating a substantial number of now-expensive metric computations.

As an alternative to MBR decoding, we can opt for using a quality-aware decoding strategy, generating a list of candidate translations and reranking them using a neural quality estimation (QE) metric that

computes a quality score conditioned only on the source and translation. This method offers the advantage of being more efficient than MBR decoding, as its inference speed scales linearly with the number of candidate translations. A study conducted by Fernandes et al. (2022) showed that employing neural metrics for QE reranking not only results in superior translations compared to beam search but also exhibits comparable advantages to those seen with MBR decoding. However, this approach still demands the use of a separate QE model to evaluate the quality of each candidate.

In our work, we propose a novel method that moves quality awareness inside the model itself, enabling the simultaneous prediction of translations and their quality scores using a single translation model, thereby obviating the need for an additional QE model during decoding. Specifically, we investigate two key strategies: 1. Quality-Aware Prompting, where we use quality prompts that explicitly encourage the generation of high-quality translations, and 2. Quality-Aware Prediction, where we enable an NMT model to judge the quality of its own translations. Both strategies add special quality tokens to each NMT training example. The strategies differ only in whether the token is included as a suffix of the source-language input, or as a suffix of the target-language output.

Our scientific contributions are:

- We introduce quality-aware translation models, demonstrating their ability to distinguish between varying qualities of their output.

- We propose a novel reranking approach that eliminates the necessity for external QE models during decoding.

- We demonstrate that our quality-aware methods notably enhance the performance of NMT models and substantially boost the efficiency of MBR decoding.

## 2 RELATED WORK

Machine translation metrics can be divided into two high-level categories: reference-based and reference-free, also known as quality estimation (QE) metrics. QE metrics compute a quality score conditioned on the source text and the MT output alone. Reference-based metrics, on the other hand, require a human-generated reference translation and primarily condition their scores on the human translation and the MT output. A multitude of reference-based metrics are available to evaluate the quality of translated content. Some metrics rely on lexical overlap, such as BLEU (Papineni et al., 2002), Meteor (Banerjee & Lavie, 2005) or ChrF (Popović, 2015). The WMT metrics task (Freitag et al., 2022b) demonstrated that the new generation of metrics – neural fine-tuned metrics like BLEURT (Sellam et al., 2020) and COMET (Rei et al., 2020a) – have significantly higher correlation with human judgement than traditional word overlap metrics. Consequently, we focus on neural fine-tuned metrics in this study. Quality estimation for MT began as confidence estimation (Blatz et al., 2004), but has recently shifted to embrace close kinship with reference-based metrics, with recent neural examples including OpenKiwi (Kepler et al., 2019), TransQuest (Ranasinghe et al., 2020), and COMET-QE (Rei et al., 2020b),

A prominent example of a neural fine-tuned reference-based oracle regression metrics is BLEURT (Sellam et al., 2020), and its extension to MetricX, which was the winning entry in the WMT22 metrics task (Freitag et al., 2022b). Building upon this foundation Anonymous (2023)[1] recently introduced BLEURT-QE, an adapted QE metric that uses the source segment as input instead of the reference to yield scores that are aligned with human preferences. In this study, we mainly focus on BLEURT as the reference-based metric and BLEURT-QE as the QE metric.

In this study, we generate a list of candidate translations by sampling, and then we use either MetricX (via MBR decoding) or BLEURT-QE to rerank the candidate list. Reranking has a long history in translation, starting with Shen et al. (2004), where a discriminative model is learned to rank a candidate list to maximize a reference-based metric, with recent examples including Bhattacharyya et al. (2021) and Lee et al. (2021), who both train using BLEU. Our approach is closest to that of Fernandes et al. (2022), who rerank using various translation quality-estimation metrics, as opposed to training a special-purpose discriminative reranker. We differ from these works in that we do not need any external quality signal, which is instead provided by the NMT system itself.

---

[1]Parts of this paper are based on work reported in papers currently under review.

While conventional MT research often relies on MAP decoding or generating k-best lists through beam search for MBR decoding, Eikema & Aziz (2020) proposed an approximation of MBR decoding via unbiased sampling. Their method aims to address the limitations of MAP decoding (Eikema & Aziz, 2020; Müller & Sennrich, 2021; Eikema & Aziz, 2022) by demonstrating that samples drawn from the NMT model align more faithfully with training data statistics when compared to beam search. In the context of applying MBR to NMT, the focus has traditionally revolved around maximizing metrics based on overlap, such as BLEU, Meteor, or ChrF, as well as their linear combinations like BEER (Stanojević & Sima'an, 2014). Freitag et al. (2022a) showed that using neural metrics instead of overlap metrics results in significant improvements in translation quality. As a follow up, Freitag et al. (2023) showed that the choice of sampling approach is important and epsilon sampling (Hewitt et al., 2022) is ideal for MBR decoding and reranking.

Our quality-aware prompting approach extends a long line of methods where tagged training data has been used to control NMT output for different properties, including target language (Johnson et al., 2016), formality level (Yamagishi et al., 2016), politeness (Sennrich et al., 2016), domain (Kobus et al., 2017), gender (Vanmassenhove et al., 2018), syntactic structure (Shu et al., 2019), complexity (Agrawal & Carpuat, 2019) and reading level (Marchisio et al., 2019). The approaches closest to ours identify attributes related to translation quality, such as tagging back-translated examples to control away from synthetic data (Caswell et al., 2019), or tagging target-original examples to control toward natural-sounding output (Freitag et al., 2022c). To the best of our knowledge, we are the first to use quality estimation to tag training data, allowing NMT to discriminate between different translation qualities, and allowing us to prompt the model to generate high quality translations.

## 3 METHOD

We are given a NMT model $P_\Theta(y|x)$ which serves to estimate the probability of a hypothesis segment $y \in \mathcal{Y}$, given a source segment $x$. Here, $\Theta$ denotes the learned parameters of the neural network and $\mathcal{Y}$ the set of all possible hypotheses. There are two widely used approaches for generating the translations of a given sentence.

**MAP decoding:** This method involves searching for the most probable translation under $P_\Theta(y|x)$. However, determining the hypothesis with the maximum probability is computationally intractable due to the expansive and combinatorially complex search space $\mathcal{Y}$. Consequently, approximations like beam search (Graves, 2012; Sutskever et al., 2014) are often employed.

**Sampling:** In many applications we want to generate diverse hypothesis such as in generative tasks where creativity is desired. In this case, instead of selecting the candidate with the highest probability (or an approximation thereof), we sample the output sentence following the probability distribution defined by the model. For MT, this approach is used for generating a list of candidate translations, e.g. for MBR decoding. Specifically, epsilon sampling, as outlined by Hewitt et al. (2022), has emerged as the leading sampling technique for MBR, as illustrated by Freitag et al. (2023) over other methods such as ancestral, top-k or nucleus sampling (Holtzman et al., 2020). Epsilon sampling prunes away any token with a probability lower than a given threshold $\varepsilon$, thereby guaranteeing that each token within a sample is allocated a fair probability mass. The likelihood of selecting token $y^{(\tau)}$ in the sampling process at time $\tau$ is governed by

$$P'_{\Theta,\varepsilon}(y^{(\tau)}|x, y^{(1:\tau-1)}) \sim \begin{cases} P_\Theta(y^{(\tau)}|x, y^{(1:\tau-1)})^{\frac{1}{T}} & \text{if } P_\Theta(y^{(\tau)}|x, y^{(1:\tau-1)}) \geq \varepsilon \\ 0 & \text{otherwise} \end{cases} \quad (1)$$

where $T$ denotes the sampling temperature. Epsilon sampling proves to be a highly effective strategy for the selective removal of unreliable, low-probability tokens.

### 3.1 EXTERNAL QE-RERANKING

External QE-Reranking involves generating a candidate list of size $N$ through sampling and then reordering these samples, according to an quality estimation (QE) model. In our experiments, we employ BLEURT-QE (Anonymous, 2023), a modification of MetricX, to compute a quality score

$q = f(s, y)$. Here, $f$ is parameterized by a transformer-based neural network and $s$ and $y$ denote the source segment and the translation, respectively. This QE approach can also be used to select not only the top-1 sample from the reranked candidate list, but also the top-k samples serving as a pre-selection task prior to MBR decoding.

## 3.2 MINIMUM BAYES RISK DECODING

In MBR decoding (Bickel & Doksum, 1977; Berger, 1985), given a set of candidate hypotheses $\mathcal{Y}$, the goal is to select the optimal hypothesis based on its expected utility concerning the distribution over human references within the space of all references $\Omega$. This can be expressed mathematically as:

$$h^{\text{best}} = \underset{y \in \mathcal{Y}}{\arg\max} \sum_{r \in \Omega} u(y, r) P_{\text{human}}(r|x)$$

where $u(y, r)$ is a utility metric that is being used to gauge the quality of a candidate translation $y$ with respect to a reference translation $r$.

Since $P_{\text{human}}(r|x)$ remains unknown, we resort to sampling from the model instead, which relies on the assumption that the model provides a reliable approximation for the true underlying distribution over human translations. Furthermore, the integration over the vast space of all possible references $\Omega$ is computationally intractable. Therefore, MBR adopts a finite sample estimate by sampling a set of pseudo-references $\mathcal{Y}_{\text{model}}$ from $P_{\text{model}}(\cdot|x)$. This approximation can be expressed as:

$$h^{\text{MBR}} = \underset{y \in \mathcal{Y}}{\arg\max} \frac{1}{|\mathcal{Y}_{\text{model}}|} \sum_{r \in \mathcal{Y}_{\text{model}}} u(y, r) \tag{2}$$

where $\mathcal{Y} = \mathcal{Y}_{\text{model}}$, as the same set of model hypotheses serves both as the candidate list $\mathcal{Y}$ as well as the pseudo-reference list $\mathcal{Y}_{\text{model}}$. The computational time complexity of MBR decoding is $O(M^2)$ with $M$ the size of the candidate list.

Note that this quadratic expression refers to *each sentence* to translate, i.e. for a corpus of size $S$, the total cost will be $O(S \cdot M^2)$. Also there is a hidden (multiplicative) constant, namely the cost of the computation of the utility function. For surface level metrics (e.g. BLEU, ChrF), this cost is negligible, but for neural metrics it involves computing the forward pass of a large neural network, thus every reduction in metric computations has an important effect on the total running cost. In this paper, we focus on using BLEURT as utility function during MBR decoding.

## 3.3 QUALITY-AWARE MODEL

In contrast to the two methods described above, which use a separate QE model for assessing the quality of translations, we propose a novel method that integrates quality awareness directly into the translation model, making an independent QE model unnecessary during decoding. We present two approaches, in the first one we prompt the model to produce translations with a high QE score. In the second one, the model is designed to provide a quality score alongside the translation. To achieve this we initially assess the quality of samples within the training dataset, employing BLEURT-QE. In the training phase we train our NMT model simultaneously on source and target samples, as well as their associated quality scores.

### 3.3.1 ASSESSING QUALITY

We first prepare the training dataset by computing the translation quality of each training sample and labelling each sentence pair with the corresponding quality score. In order to ease the task of the model we discretize the scores via equal mass binning into $B$ bins, which are then mapped to single tokens of the vocabulary of the translation model. To achieve this, we consider the set of quality scores $Q$ from all samples in the training dataset, denoted as $q_1, q_2, \ldots, q_N$, in order to determine bin boundaries or cut-off points $c_1, c_2, \ldots, c_{B+1}$ such that:

$$|\{x \mid q \in Q, \ c_i \leq q < c_{i+1}\}| \approx \frac{N}{B} \qquad \forall 1 \leq i \leq B \tag{3}$$

In this way each bin contains approximately the same number of samples. This is to avoid sample imbalances per bin when training the model. Next, each bin is assigned a bin identifier $b$, which is represented by a single token in the model vocabulary. E.g. if we define 10 bins, we can just use the numbers between 0 and 9. Lastly, the quality score is inserted into the data pipeline during training to associate the source and target pair with the respective bin identifier $b$. To mark the token $b$ as a QE value, we employ a special string format by surrounding $b$ with square brackets: $[b]$. In the following we outline two distinct methods for integrating our quality score string into the model.

### 3.3.2 QUALITY-AWARE PROMPTING (QA PROMPTING)

During the training process, we append the quality score string $[b]$ as a suffix to the source segment (Figure 5a). This approach enables the model to learn to associate the given discretized quality score $b$ with the source $s$ and target $t$ segments. As the quality token is attached to the *input* to the system, it also provides us with the capability to prompt for high-quality translations directly during decoding. I.e. at translation time we append the token corresponding to the highest quality level to the sentence to translate to prompt the system to generate a sentence of the highest quality.[2]

### 3.3.3 QUALITY-AWARE PREDICTION (QA PREDICTION)

Instead of prompting the model explicitly for high-quality translations, an alternative approach is to design a model that jointly predicts a hypothesis and a quality score. This design allows us to leverage our translation model to also function as a QE model. To achieve this, we append the quality score string $[b]$ to the target sentence during training, enabling the model to learn to predict the quality during inference (Figure 5b).

If using a reranking approach these quality scores can be directly utilized to reorder samples within a candidate list. However, due to our use of discretized bins, it is possible that the model predicts the same scores for multiple samples. To address this, we also consider the log probabilities $z$ associated with the bin identifier tokens. This additional information allows for a more precise reranking of samples in the candidate list. Specifically, we sort samples with respect to $b$ as the primary sorting criterion and use the log probabilities $z$ as the secondary criterion. Given a candidate list of size $M$, we sort the samples into $y_1, y_2, \ldots, y_M$ with corresponding discretized quality scores $b_1, b_2, \ldots, b_M$ and log probabilities $z_1, z_2, \ldots, z_M$ in such manner that

$$\forall 1 \leq i < j \leq M : (b_i > b_j) \text{ or } (b_i = b_j \text{ and } z_i > z_j) \tag{4}$$

With the sorted candidate list in hand, we can proceed by either selecting the top-ranked sample as our final translation or further processing the top-k samples in the context of MBR decoding.

## 4 EXPERIMENTAL SETUP

**Model:** Our model is based on a transformer architecture, consisting of 6 encoder and 6 decoder layers, 16 attention heads with a dimension of 128, a hidden dimension of 8192, and a model dimension of 1024, resulting in 551M parameters. In our experiments, we employ a shared vocabulary of 32k tokens and impose a maximum sentence length of 128 tokens. We utilize GELUs with gated activation functions. The baseline system is trained on the entire available dataset. All models are trained until they reach convergence. The BLEURT-QE model used for quality estimation is a transformer based model with a total of 2B parameters, as described in Anonymous (2023). To assess the applicability of our approach to LLMs, we also train and evaluate a LLM model and show the results in the appendix Tab. 3.

**Data:** We choose two high-resource language pairs from the WMT 2022 shared task: English to German (en → de) and in the Appendix we additionally show results for English to Japanese (en → ja). While we filter out sentences exceeding 128 tokens, we perform no further data filtering or preprocessing. The training dataset for (en → de) comprises 2955.8M samples, while the (en → ja)

---

[2]Of course we can prompt with any of the quality levels, but we are only interested in the best quality translations.

dataset consists of 33.9M samples. Our evaluation is based on the WMT 2022 general translation task test sets.

Given that we have available BLEURT-QE scores for the whole training data, one natural question to ask is what would happen if we limit the training data to the only the best quality training sentence pairs. (Anonymous, 2023) showed that this is indeed a very effective way to reduce the training data size, while at the same time improving translation performance. We also report experiments on this data condition, which represents a stronger baseline with which to compare our methods. For these experiments we follow (Anonymous, 2023) and keep only the top 50% scoring sentence pairs.

**Metrics:** We use neural metrics for evaluation, with a focus on COMET (Rei et al., 2020a) (COMET 22 version). We also report MetricX scores, but as BLEURT-QE is based on it and our methods directly optimize this metric, there is the danger of overfitting for this particular metric (Amrhein & Sennrich, 2022; Yan et al., 2023). In addition, for selected experiments we conducted additional expert-based human evaluations using MQM (Freitag et al., 2021), a human evaluation scheme centered on marking errors present in the translations.

## 5 RESULTS

### 5.1 CAN LMS DISTINGUISH BETWEEN THE QUALITY OF THEIR OUTPUTS?

In our first experiment, we aim to investigate whether our proposed approach can effectively instill quality awareness into NMT models. To this end, we assess the model's capability to accurately predict quality scores of translations and, crucially, its ability to differentiate between different quality levels. We train a model using the proposed Quality-Aware Prediction method, where the model's objective is to predict both a translation and a (discretized) quality score. We then compare the predicted quality scores with the ground truth scores calculated using the BLEURT-QE model for each source and translation pair.

Figure 1a shows a histogram with the distribution of the ground truth QE scores[3] through the different predicted quality bins. We observe that the model, while slightly overoptimistic, performs well in assigning samples with high ground truth values to the highest quality score bin, and vice versa. Additionally, the predicted quality score bins are well aligned with the ground truth scores, as demonstrated in Figure 1b, where we show the distribution of ground truth scores (non discretized) across the predicted bins. The correlation coefficients between the ground truth QE scores and our model's predicted scores are as follows: 0.73 for Pearson, 0.72 for Spearman's rank, and 0.58 for Kendall's Tau correlation. To measure the correlation coefficients, we assign a score to each bin, which represents the mean QE score in a given bin based on the training dataset.

### 5.2 TRANSLATION PERFORMANCE

Once we confirmed that our model is indeed able to distinguish between qualities regarding its own outputs, we explore how to utilize this property to enhance the overall output quality of a NMT system. For this experiment, we evaluate both our Quality-Aware Prediction model and Quality-Aware Prompting model.

For the Quality-Aware Prediction model, we extract $N$=1024 hypotheses from the model via epsilon sampling and retrieve the quality score string suffix from each hypothesis. Subsequently, we rank these hypotheses using Equation 4 and select the highest-ranked sample as our final translation output. In the case of Quality-Aware Prompting, we directly retrieve the final translation through MAP-decoding by appending the highest quality score string as a suffix to the source sentence, i.e. we "ask the model" to produce high quality outputs. We compare these two approaches with an identical baseline NMT model that only differs by not using any quality score during training.

Table 1 demonstrates that both quality-aware methods surpass the baseline model in terms of MetricX and COMET, with QA Prompting showing better results on both metrics. As expected, all the methods achieve big improvements in MetricX, as it is closely related to the BLEURT-QE metric that we

---

[3]Note that the ground-truth scores have been discretized taking into account the range present in the development set.

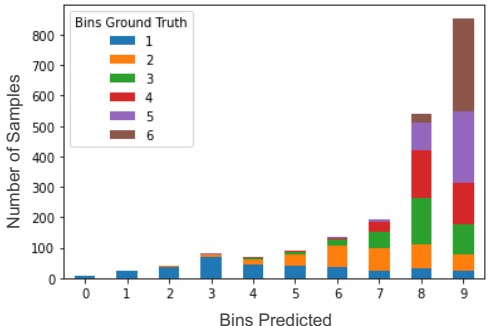

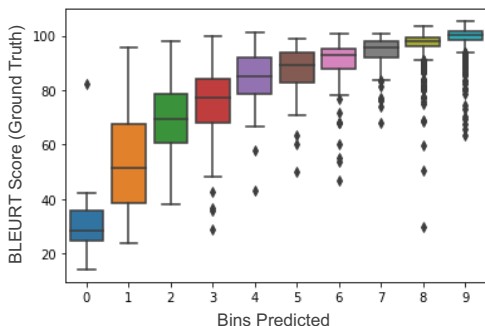

(a) Histogram over predicted score bins with colors indicating ground truth qualities. Samples with high ground truth scores are mainly assigned to the highest predicted bins and vice versa.

(b) Boxplots that show the distribution of ground truth scores across all samples assigned to each bin. The median ground truth quality score increases steadily in line with the predicted bins.

Figure 1: Alignment between predicted quality scores from proposed Quality-Aware Prediction model and ground truth BLEURT-QE scores of translations in the en → de test dataset.

Table 1: Comparison between quality-aware models and baseline models on the full and filtered training datasets. The quality-aware methods outperform the baseline model and perform similar to reranking without requiring an additional BLEURT-QE model during decoding. ⋆ denotes statistically significant (pairwise permutation test Koehn (2004) with p=0.05) differences compared to the baseline with $p < 0.05$. No significance is computed for MetricX due to the methods optimizing this metric directly.

| Method | Data | MetricX | COMET | MQM ↓ |
|---|---|---|---|---|
| Baseline | Full | 80.2 | 85.8 | 2.402 |
| QA Prompting (Ours) | Full | **82.3** | **87.1**⋆ | **1.855**⋆ |
| QA Prediction (Ours) | Full | 82.0 | 86.5⋆ | 2.796 |
| External QE-Reranking | Full | 83.3 | 86.9⋆ | 1.889⋆ |
| Baseline | Filter | 81.8 | 87.0 | – |
| QA Prompting (Ours) | Filter | **82.6** | **87.3**⋆ | – |
| QA Prediction (Ours) | Filter | 82.5 | 86.7 | – |
| External QE-Reranking | Filter | 83.7 | 86.9 | – |

are directly optimizing. In fact, external QE-Reranking achieves the best MetricX score by a wide margin, however COMET puts QA Prompting on-par with QE-Reranking.

The human evaluation majorly confirms the trends showed by COMET. The MQM scores can be interpreted as the average number of errors in translation, i.e. lower numbers are better. The human evaluation shows that QA Prompting does indeed produce significantly better translations than the baseline systems, and it comes close to the external QE-Reranking approach (the difference is not statistically significant). This is specially noteworthy given that QA Prompting is a single pass approach, with no additional cost over the MAP decoding baseline, whereas QE-Reranking rescores a 1024 candidate list with an external (expensive) QE model. Regretfully, QA Prediction did not outperform the baseline approach and in fact showed a degradation in performance.

When compared to the stronger baseline where the training data is prefiltered with BLEURT-QE scores, we see that QA-prompting is still able to obtain a slight improvement over both the baseline and QE-Reranking, which is still statistically significant. QA-prediction is not effective in this setting.

## 5.3 MBR DECODING

Next we turn our attention to improving the performance of MBR decoding. This depends heavily on the length of the candidate list $M$, and Freitag et al. (2023) showed that a large candidate size

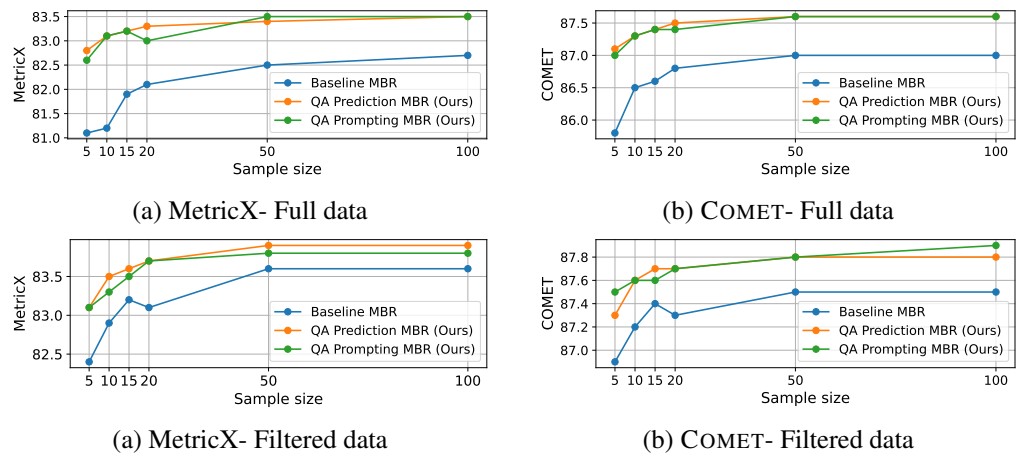

(a) MetricX- Full data

(b) COMET- Full data

(a) MetricX- Filtered data

(b) COMET- Filtered data

Figure 2: Performance of quality-aware approaches (Quality-Aware Prediction and Quality-Aware Prompting) compared to baseline MBR decoding across various candidate list sizes. MBR decoding with Quality-aware models consistently outperforms baseline MBR decoding across candidate list sizes. The quality-aware approaches can achieve the same level of performance as baseline approaches while reducing the required utility function computations by one to two orders of magnitude.

of several hundred candidates is needed for achieving good translation performance. However, this property makes MBR decoding computationally expensive as the utility function computations grow quadratically with the candidate list size $M$, see Section 3.2. Our investigation seeks to understand how improved candidate quality influences the performance and efficiency of MBR decoding.

For baseline MBR decoding we use epsilon sampling to generate a candidate list of $M$ samples, similarly for the Quality Aware prompting approach. For Quality-Aware Prediction, in line with the previous section, we employ our Quality-Aware Prediction approach to sample $N = 1024$ hypotheses. Then we rank all samples and select the top $M$ samples as our candidate list. Subsequently, MBR decoding is applied to the gathered candidate lists. We used BLEURT as utility function as a proxy for MetricX, due to the high computational cost of this last metric.

In Figure 2 we show the performance of our quality-aware approaches compared to baseline MBR decoding across various candidate list sizes $M$. Our proposed methods consistently outperform baseline MBR decoding in terms of MetricX and COMET scores, irrespective of the candidate list size. Notably, our quality-aware approaches combined with MBR decoding require substantially fewer candidates to achieve equivalent performance to baseline MBR decoding. For example, in models trained on the entire dataset, the Quality-Aware Prompting and Quality-Aware Prediction approaches obtain COMET scores of 86.9 and 87.0, respectively, with a candidate list size of 5 (with just 20 utility function computations[4] per sentence). In contrast, baseline MBR decoding plateaus at 87 COMET starting at a candidate size of 50 (requiring 2450 utility function computations per sentence). This translates to a 100-fold increase in computations for the baseline model to achieve a similar score. We also note that the baseline model with a candidate list size of 1024 achieves a MetricX score of 82.9 and a COMET score of 87.0. This indicates that our approach, with a candidate list size of 50, outperforms a baseline model with even 1024 samples. For our experiments on filtered data we observe a similar improvement in the quality-aware models when compared to baseline MBR decoding.

Table 2 shows the translation performance of the MBR systems, including human evaluation with MQM, with a candidate size $M = 50$. A first observation is that baseline MBR decoding significantly outperforms the baseline system, coming close to the external QE reranking approach (the difference to it is not significant). When we combine MBR with quality-aware models, we again obtain a significant improvement when compared the the MBR baseline, with the MQM score dropping from 1.861 to 1.542 for QA prompting. In this condition, the QA prediction approach does perform satisfactorily, and there is no significant difference when compared to QA prompting.

---

[4]The number of computations is $M \times (M - 1)$ as a hypothesis is not evaluated against itself.

Table 2: MBR results with Quality-Aware decoding approached. $^\star$ denotes statistically significant differences compared to the baseline with $p < 0.05$, $^\dagger$ denotes statistically significant differences compared to the *MBR baseline with* $M = 50$. No significance is computed for MetricX due to the methods optimizing this metric directly.

| Data | #Candidates | Method | MetricX | COMET | MQM |
|---|---|---|---|---|---|
| Full Corpus | | Baseline (w/o MBR) | 80.2 | 85.8 | 2.402 |
| | | External QE-Reranking | 83.3 | 86.9$^\star$ | 1.855$^\star$ |
| | $M = 50$ | MBR Baseline | 82.5 | 86.8$^\star$ | 1.861$^\star$ |
| | | MBR QA Prompting (Ours) | **83.5** | 87.4$^{\star\dagger}$ | **1.542**$^{\star\dagger}$ |
| | | MBR QA Prediction (Ours) | 83.4 | **87.5**$^{\star\dagger}$ | 1.555$^{\star\dagger}$ |
| | $M = 5$ | MBR Baseline | 81.1 | 85.7 | – |
| | | MBR QA Prompting (Ours) | 82.6 | 86.9$^\star$ | – |
| | | MBR QA Prediction (Ours) | **82.8** | **87.0**$^\star$ | – |
| Filtered | | Baseline (w/o MBR) | 81.8 | 87.0 | – |
| | | External QE-Reranking | 83.7 | 86.9 | – |
| | $M = 50$ | MBR Baseline | 83.6 | 87.4 | – |
| | | MBR QA Prompting (Ours) | 83.8 | 87.7$^{\star\dagger}$ | – |
| | | MBR QA Prediction (Ours) | **83.9** | 87.7$^{\star\dagger}$ | – |
| | $M = 5$ | MBR Baseline | 82.4 | 86.7 | – |
| | | MBR QA Prompting (Ours) | **83.1** | **87.3**$^\star$ | – |
| | | MBR QA Prediction (Ours) | **83.1** | 87.2$^\star$ | – |

We can even be more aggresive, and reduce the size of the candidate even further to $M = 5$, as shown in Table 2. We can see that the translation performance drops only slightly (e.g. only 0.3 COMET for the Filtered QA Prompting approach), but the number of utility function computations is drastically reduced from 2450 to 20, two orders of magnitude. This is not the case for baseline MBR, where with this reduced candidate size, even performs below the non-MBR baseline.

## 6 CONCLUSION

This paper introduces a novel approach to enhance NMT by making the models quality-aware. Our approach addresses the issue of misalignment between outputs generated via MAP decoding and human judgment. We achieve this by training NMT models to self-assess the quality of their own translations, effectively circumventing the limitations of conventional decoding methods. As a result, this new approach yields significantly improved translation quality, outperforming quality-aware decoding strategies like QE-reranking. Using the prompting strategy, the improvements come at no additional computational cost, as only a single MAP decoding pass is needed. QE-reranking in contrast needs a sampling step followed by a reranking step using an external, computationally expensive additional model.

By leveraging the model's quality signal internally, not only does translation quality further improve, but computational efficiency is also dramatically enhanced, reducing inference time by two orders of magnitude compared to state-of-the-art MBR decoding. Furthermore, we demonstrate in the appendix that also LLMs can be effectively made quality-aware through finetuning using our method, thereby benefiting from the same advantages as Encoder-Decoder models trained from scratch. This research opens up exciting possibilities for advancing the field of NMT, offering both improved translation quality and faster processing speeds without the need for additional, computationally intensive models.

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

## A  QUALITY AWARE LLMS

Table 3: Quality Aware LLMs: The performance of LLMs can be enhanced via finetuning a pretrained LLM with Quality-Aware Prompting compared to standard finetuning.

| Method | BLEURT | BLEU |
|---|---|---|
| LLM Baseline | 77.9 | 36.6 |
| LLM Quality-Aware Prompting (Ours) | **79.2** | **37.3** |

In this section we explore the potential of enhancing LLMs with quality awareness through our proposed method. Specifically, we employ our most efficient approach, Quality-Aware Prompting, to finetune a LLM[5]. To this end, we finetune the pretrained model for 10k steps using our Quality-Aware Prompting approach. As baseline we also finetune the LLM with identical configurations on the same data, but without incorporating any quality signal. In Table 3, we observe that Quality-Aware Prompting substantially outperforms standard finetuning by 1.3 BLEURT points. Our results suggest that substantial performance improvements can be achieved with minimal data and steps. This opens the door to the possibility of utilizing more costly QE methods or even human evaluations in the future to curate finetuning datasets and align models directly with human preferences.

## B  ADDITIONAL RESULTS FOR LANGUAGE PAIR: ENGLISH TO JAPANESE

Beyond the results highlighted in the main text, we present findings for an additional language pair, specifically, English to Japanese ($en \rightarrow ja$). Our experimental setup mirrors that of the English to German translation task, with the exception that we employ a BLEURT score threshold of 60 for training data filtering. Our results closely resemble those obtained for the English to German datasets.

Exploring the $en \rightarrow ja$ scenario, we compare the performance of quality-aware models against baseline models using both the complete and filtered training datasets. Evaluation metrics include MetricX, BLEURT, COMET, and BLEU. Notably, the quality-aware methods achieve consistently better results than the baseline model, all without the need for an additional BLEURT model during decoding (Tab. 4).

In Fig. 3 we also examine the effectiveness of the proposed approaches in contrast to baseline MBR decoding across various candidate list sizes for $en \rightarrow ja$. Our findings demonstrate that MBR decoding with quality-aware models consistently surpasses baseline MBR decoding across different candidate list sizes. When trained on the entire dataset our methods achieve MetricX and COMET scores with only 5 samples that are clearly better than the baseline model regardless of the sample size. For this language pair we can dramatically increase the performance while at the same time requiring more than two orders of magnitude less computation time. When trained on the full dataset, our methods exhibit MetricX and COMET scores that outperform the baseline model for all candidate list sizes with a noticeable advantage, even when considering a limited number of samples. Importantly, this improved performance is accompanied by a similar reduction in computation time as for $en \rightarrow de$, with our approach requiring two orders of magnitude less computational resources for this language pair.

## C  SENSITIVITY ANALYSIS OF NUMBER OF BINS

In this section, we examine the selection of the number of bins for discretizing the quality score. To do this, we employ our Quality-Aware Prediction approach and train five models with varying numbers of bins, specifically 2, 3, 5, 10, and 20. We then evaluate their performance on MetricX, BLEURT, COMET, and BLEU. Figure 4 illustrates that increasing the number of bins yields improvements on our quality metrics, in particular in the range of 2 to 5 bins. This aligns with our overarching concept of instilling quality awareness in the model, as a higher number of bins allows for a finer distinction between quality levels within the model, which is evident in our findings.

---

[5]We keep the model anonymous for reviewers and add references to the model in the camera ready version.

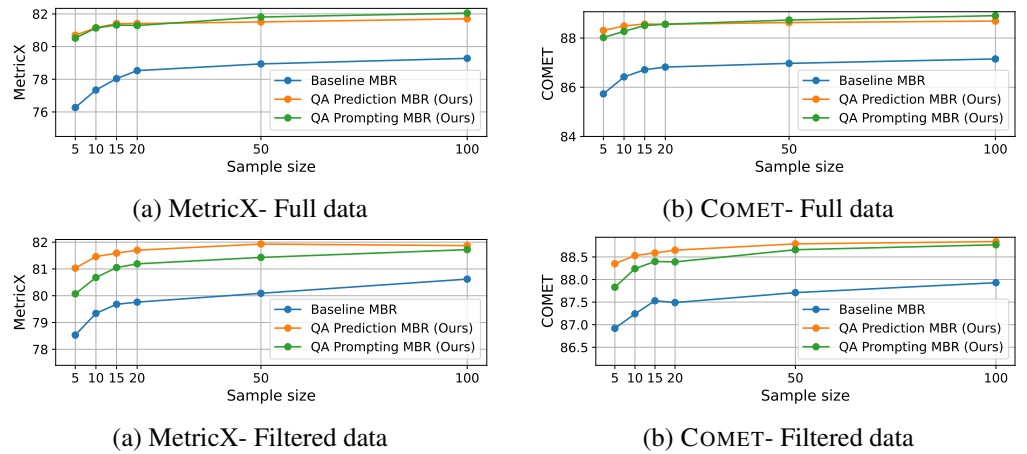

(a) MetricX- Full data

(b) COMET- Full data

(a) MetricX- Filtered data

(b) COMET- Filtered data

Figure 3: Performance of quality-aware approaches (Quality-Aware Prediction and Quality-Aware Prompting) compared to baseline MBR decoding across various candidate list sizes for $en \to ja$. MBR decoding with Quality-aware models consistently outperforms baseline MBR decoding across candidate list sizes. The quality-aware approaches can achieve the same level of performance as baseline approaches while reducing the required utility function computations by one to two orders of magnitude.

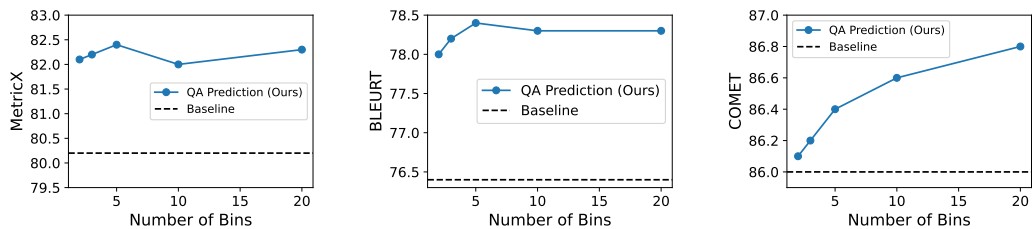

Figure 4: Sensitivity concerning performance of the Quality-Aware Prediction approach w.r.t. the number of quality score bins. Increasing the number of quality score bins yields generally improvements on our quality metrics, specifically in the range of 2 to 5 bins.

Table 4: Comparison $en \rightarrow ja$ between quality-aware models (Quality-Aware Prediction and Quality-Aware Prompting) and baseline models on the full and filtered training dataset evaluated on MetricX, BLEURT, COMET and BLEU. The quality-aware methods outperform the baseline model without the need of an additional BLEURT model during decoding.

| Method | Data | MetricX | COMET |
|---|---|---|---|
| Baseline | Full | 76.3 | 85.7 |
| Quality-Aware Prompting (Ours) | Full | 80.3 | **87.8** |
| Quality-Aware Prediction (Ours) | Full | **80.9** | 87.7 |
| Baseline | Filter | 77.8 | 86.2 |
| Quality-Aware Prompting (Ours) | Filter | **80.0** | **87.7** |
| Quality-Aware Prediction (Ours) | Filter | 79.6 | 87.5 |

## D  INFLUENCE OF BIN IDENTIFIER CHOICE

Tokens in a language model are mapped to an embedding, representing a specific meaning or relation to other tokens within the embedding space. In this context, we aim to explore whether mixing token meanings from the primary translation task with the scoring task has a detrimental effect on either one of them. To investigate this, we assess the translation quality of the baseline model without the inclusion of magic quality score strings, as well as our proposed Quality-Aware Prediction approach.

Firstly, we evaluate the translation quality between the baseline and our model. We find that our model performs equivalently to the baseline when treating it as a basic translation model and disregarding the magic quality score strings appended to our model's hypotheses. This suggests that the model successfully disentangles a token's actual meaning in the text from its role as a quality score bin identifier.

Furthermore, we investigate whether the use of tokens that appear frequently in the training data corpus, such as numbers and letters, as opposed to tokens that are the least likely to appear in the training data, has an adverse impact on quality scoring. To investigate this, we train our model with 5 bins but employ different bin identifiers, including numbers $[0, 1, 2, 3, 4]$, letters $[a, b, c, d, e]$, and the 5 least frequently occurring tokens from the vocabulary. Tab. 5 shows that varying the choice of bin identifiers demonstrates a high degree of robustness.

Table 5: Sensitivity of our Quality-Aware Prediction approach with respect to bin identifiers on the entire training dataset. For all experiments we set the number of bins to 5 and vary the choice of bin identifiers. Our approach is largely insensitive to bin identifiers.

| Bin identifiers | MetricX | COMET |
|---|---|---|
| Baseline | 80.2 | 86.0 |
| Numbers | 82.4 | 86.4 |
| Letters | 82.5 | 86.6 |
| Least likely tokens | 82.6 | 86.6 |

## E  COMBINING QUALITY-AWARE PROMPTING AND QUALITY-AWARE PREDICTION

Throughout our experiments we frame Quality-Aware Prompting and Quality-Aware Prediction as two separate approaches. One might wonder whether both approaches are orthogonal to each other and might benefit each other when combined. To this end we add the magic quality score string to the source and the target sentence. To avoid that the model just learns copying the magic string from the input to the output, we choose a multiple of the bin number from the prompting approach for the prediction approach. This way we make sure that the model is required to first of all learn to provide a high quality translation when prompted for it and learns to fine grained distinguish the quality in the

quaility prediction tasks in the output. However, we observe that the combination of both approaches results in inferior performance compared to each approach individually, regardless of whether the full or filtered dataset is employed for training (Tab. 6). We hypothesize that this may be attributed to the model becoming excessively fixated on predicting the score in the output based on the input score, potentially leading to overfitting, where the prediction score becomes overly conditioned on the prompting score.

Table 6: Performance of combining Quality-Aware Prompting and Quality Prediciton approaches on both full and filtered datasets. Two combined models are trained for each dataset: one with 10 and the other with 20 prediction quality score bins, while using 5 quality score bins for prompting. Combining both approaches yields no improvements across metrics.

| Method | Data | MetricX | COMET |
|---|---|---|---|
| Quality-Aware Prompting | Full | 82.3 | 87.1 |
| Quality-Aware Prediction | Full | 82.0 | 86.6 |
| Combo: ProBins=5 and PreBins=10 | Full | 82.1 | 86.7 |
| Combo: ProBins=5 and PreBins=20 | Full | 82.1 | 86.7 |
| Quality-Aware Prompting | Filter | 82.6 | 87.3 |
| Quality-Aware Prediction | Filter | 82.5 | 86.9 |
| Combo: ProBins=5 and PreBins=10 | Filter | 81.9 | 86.6 |
| Combo: ProBins=5 and PreBins=20 | Filter | 82.1 | 86.8 |

## F  DETAILS OF QUALITY-AWARE APPROACHES

In the context of Quality-Aware Translation, two approaches are explored. In Quality-Aware Prompting (QA Prompting), a quality score is added as a suffix to the source segment during training, facilitating the model's association of quality scores with source and target segments (Figure 5a). This appended quality token in the input allows for direct prompting of high-quality translations during decoding. Conversely, Quality-Aware Prediction (QA Prediction) involves training a model that predicts both a hypothesis and a quality score concurrently. This approach transforms the translation model into a Quality Estimation (QE) model by appending the quality score string to the target sentence during training, enabling the model to predict quality during inference (Figure 5b).

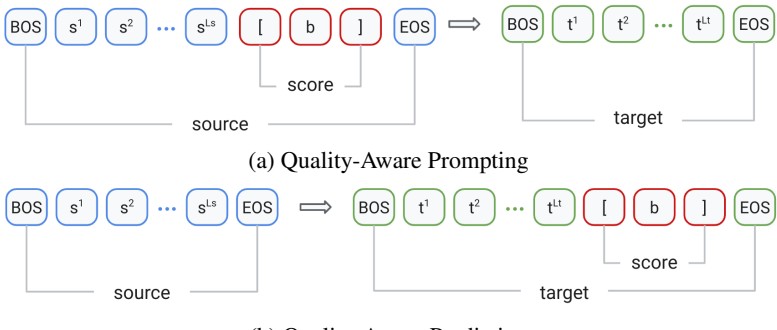

(a) Quality-Aware Prompting

(b) Quality-Aware Prediction

Figure 5: Visualization of the Quality-Aware Prompting and Quality-Aware Prediction approaches.

