# OpenReview forum: "Quality Control at Your Fingertips: Quality-Aware Translation Models"
_ICLR.cc/2024/Conference — ICLR 2024 Conference Withdrawn Submission_

### Official Review · Reviewer_9pCm · 2023-10-22

**Soundness:** 2 fair
**Presentation:** 2 fair
**Contribution:** 2 fair
**Rating:** 3
**Confidence:** 5

**Summary:**

they propose to make the NMT models themselves quality-aware by training them to estimate the quality of their own output. During decoding, they can use the model’s own quality estimates to guide the generation process and produce the highest-quality translations possible. They demonstrate that the model can self-evaluate its own output during translation, eliminating the need for a separate quality estimation model.

**Strengths:**

- This paper trains the so-called quality-aware NMT models, which is somewhat novel.

**Weaknesses:**

- This paper is not well written, I mean, they just train the model with two tasks at the same time, i.e., translation and sentence-level QE. Their model does not have the ability to evaluate the output's quality, which is their argued contribution.
- From the training procedure, I can not get how the model is aware of the translation quality. The method and experiments are not convincing enough.
- The results listed in the experiments are not enough. Why not report the BLEU score? I guess, the BLEU score is not improved as this paper uses the BLEURT-QE as the quality estimation.

**Questions:**

- See above

---

> ### Author Response · Authors · 2023-11-22
>
> We thank the reviewer for their constructive comments and answer their questions below.
>
> 1. Would you mind elaborating on where specifically you see potential to improve the paper? We agree that the sole prediction of quality labels across all quality levels is not perfect. We also refer to the response for reviewer #2 for a short discussion about correlation with human judgement. Nevertheless, we note that QE is not the goal of this paper since we are mainly concerned with the fact that good hypotheses get good scores, and we can take advantage of that in order to choose a sample with a high quality score. Furthermore, we are reordering the the samples in the highest bucket based on the log probability of the respective score token, which results in better performance w.r.t. reference based metrics, which is our goal of this paper. We will make this objective clearer in the camera ready version.
> 2. The general idea of our approach is that we supply the quality labels at training time, the system thus learns from them. For more details please refer to the “Method” section in our submitted paper. Regarding our experiments, we evaluated our proposed methods on 2 language pairs as well as provide several metrics, including human evaluations, which show that particularly quality prompting performs superior to baseline methods (Tabel 1 & 2, Figure 1). May we ask you, what in particular you find to be “not convincing enough”?
> 3. Since BLEU is a lexical metric, it has been shown to be less reliable in judging translations compared to neural metrics or even human evaluations. Since the scientific community is in the process of transitioning away from BLEU, we did not include it in our paper. Nevertheless, we understand that BLEU has been around for a long time and this is why we will include it in the appendix in the camera ready version for the sake of completeness.

---

> > ### Comment · Reviewer_9pCm · 2023-11-23
> > **Response to the rebuttal**
> >
> > Hi, authors:
> >   Thanks for your rebuttal.  For a more fair and concise review, we re-read your paper carefully and respond your rebuttal as follows:
> >   1. I agree that the idea of quality-aware NMT is very interesting, but just appending a QE label into the source or target sentence never makes the model aware of the quality. As for the predict-prompt model in you work, if we change the QE label into the source sentence, can your model predict sentences with quality changed with the QE label? If so, please provide case studies in your test sets.
> >   2 I noticed that you test the correlation with your predicted QE label with human judgment in your rebuttal to reviewer #2, and the Peasron has reached to 0.93.  Show me your detailed setup about the testing. Have you used the DA dataset in QE?
> >   3 Enable the model to be aware of the quality of the translation. Many pioneer works have been proposed: such as GAN in NMT, MRT training in NMT, and recently the RRHF or DPO in NMT.  If your paper is focused on this topic, please cite and comapre with these papers.
> >   4 I cannot agree with you that "Since the scientific community is in the process of transitioning away from BLEU". I believe most of the reviewers or researchers cannot agree with this point. We are glad to see more metrics to be used in MT, but BLEU is still a golden standard metric in MT.

---

> ### Author Response · Authors · 2023-11-23
>
> 1. We are afraid we are hitting a terminology problem here. We showed in the paper that by querying the model with "high quality" prompts we were able to improve translation quality and/or reduce the size of the candidate list for MBR decoding. Note that these two are the main takeaways from the paper, we are not claiming to have developed an improved evaluation for MT quality or anything along these lines. In the response to reviewer SaRa, we provided evidence that our system was even able to perform QE scoring to some degree (which, again, is not the main purpose of the system). We chose the term "quality-aware" as for us it made intuitive sense and we think it provides a valid explanation for the improvement in translation quality. If that is a friction point, we can happily change the name to something less problematic.
>
> 2. We took the data from the WMT23 metrics tasks (https://wmt-metrics-task.github.io/) and for each (source, MT output) pair we force decoded with our systems until the end of the MT output. Then we let our system generate the quality label and used this as a QE estimation (to be fair, we restricted the output of our system only to the possible quality lables, in order to force it to act as a QE metric instead of extending the translation). We then evaluated with the official tools of the evaluation, the human scores used for reference are MQM-based, not DA. But we want to stress this point again: the main goal of the paper is the increase in translation quality and the decrease of MBR computational cost, not QE.
>
> 3. We probably are hitting the same problem about terminology here. But referring to the approaches you mention definitely makes sense in the scope of the paper. Our approach definitely is much more lightweight than these, and thus limited in its effect.
>
> 4. The main forum for MT metrics development and evaluation is the WMT conference, with its yearly metrics task. The findings paper of last year could not be more explicit: "Results of WMT22 Metrics Shared Task: Stop Using BLEU – Neural Metrics Are Better and More Robust" (https://aclanthology.org/2022.wmt-1.2/). The MT community certainly is shifting away from BLEU, the more general ML community should follow as well.

---

### Official Review · Reviewer_SaRa · 2023-11-01

**Soundness:** 2 fair
**Presentation:** 2 fair
**Contribution:** 2 fair
**Rating:** 3
**Confidence:** 4

**Summary:**

This paper presents methods for training an NMT model to perform quality estimation along with translation. One benefit of this paradigm is that the MBR decoding could be performed with a high efficiency. The experiments show that the translation quality could be improved.

**Strengths:**

The methods are: appending the QE label to the source segment; appending the QE label to the target segment, which are both quite easy to perform.

With the proposed method, the MBR decoding could achieve competitive performance with 10-20 candidates, which is much smaller than reported in previous paper.

**Weaknesses:**

1. generating a quality label does not necessarily mean that the model has the ability to predict it. I am wondering if there is some disturbances are made to the sentence in the training data, will the proposed model generate the correct quality label (showing the quality goes down)?

2. according to fig.1 , the prediction of quality labels is not good at all. The model seems not to be able to discriminate candidates with different qualities.

3. using QE label as the generation labels seems to be an interesting idea. Will you please give some examples of the same source sentence translated with different QE labels? It would be nice to see the effect demonstrated.

4. I am not quite sure how is the quality difference between two translations with 1 point difference in MetricX or Comet score. It will be better to give some examples to show how the translation quality is improved indeed.

**Questions:**

See the weakness part for the details.

**Details Of Ethics Concerns:**

The paper is based on one anonymous paper, presenting BLEURT-QE. The paper is under review, which is not accessible.

Sec. 3.1 use BLEURT-QE as its core method to assign quality score to each sentence of the training set, which I believe is very important to the success of the whole paper.

The attribution of the method BLEURT-QE is unclear.

The effects of BLEURT-QE is not demonstrated, which makes part of this work unclear.

---

> ### Author Response · Authors · 2023-11-22
>
> We thank the reviewer for their constructive comments and answer their questions below.
>
> 1. We tested our approach on the WMT23 QE estimation task by force decoding the given source and target sentences and letting the model generate the corresponding QE label. We obtain a Pearson system-level correlation of 0.93. Comparative data points: best QE metric 0.98, CometKiwi 0.95, BLEU 0.92, chrF 0.87 (note that the last two metrics are reference-based). Thus, although it is not the best performing QE system, our approach does learn to estimate the quality of the translations.
> 2. We agree that the sole prediction of quality labels across all quality levels is not perfect. However, this is not the goal of this paper since we are mainly concerned with the fact that good hypotheses get good scores, and we can take advantage of that in order to choose a sample with a high quality score. Furthermore, we are reordering the the samples in the highest bucket based on the log probability of the respective score token, which results in better performance w.r.t. reference based metrics, which is our goal of this paper. We will make this objective clearer in the camera ready version.
> 3. This is a great suggestion and we will include examples of translations with different labels in the paper.
> 4. We refer the reviewer to the following paper for a discussion about the interpretation of neural metrics: https://aclanthology.org/2023.mtsummit-research.16.pdf. When available, human evaluation is the main metric that we use for interpreting the results, and the results on MQM are favorable for out method. As with the last point, we will add examples to the paper. As are representative, we can add some examples here where the (German) output of the quality prompting system is much more natural than the baseline System:
>
> Baseline: Bei Bestellung ist es zur angegebenen Lieferzeit.
>
> QA prompting: Bei Bestellung wird die Lieferzeit angegeben.
>
> —
>
> Baseline: Deshalb bitten wir Sie, diese Vorlaufzeiten bitte zu erlauben.
>
> QA prompting: Deshalb bitten wir Sie, diese Vorlaufzeiten einzuplanen.
>
>
> With respect to the BLEURT-QE metric we are happy to report that the paper has been accepted for publication in the WMT conference, we refer the reviewer to that paper for a detailed explanation of the metric and analysis of the results.

---

### Official Review · Reviewer_e2tN · 2023-11-01

**Soundness:** 3 good
**Presentation:** 3 good
**Contribution:** 3 good
**Rating:** 6
**Confidence:** 4

**Summary:**

The paper proposed two methods to make the NMT model quality aware. One is to prompt the NMT model with a quality score during training, but using the best score during inference time. The other is similar to multi-task learning but in a more unified way by appending the quality score in the target side. Both approaches show promising improvements in translation quality and one of them can work well with the MBR decoding to boost the translation quality further.

**Strengths:**

The paper targets an interesting and essential problem for NMT, which is both related to the translation safety and quality. It proposes two novel and efficient methods. Both methods are very simple but effective according to the experiments. The paper is also well written and clear to me mostly. I believe the proposed methods have the potential to be applied to large scale translation systems.

**Weaknesses:**

My concerns are in the questions. If they can be addressed properly, they won't be weakness to me.

**Questions:**

In conclusion, which one between QA prompting and prediction approaches is your recommendation in the situations including latency sensitive inference and large scale distillation. Please also describe how do you scale your methods in large scale multilingual machine translation system. The experiments highly relies on the model best evaluators. How do you make a cold start on a low resource setting?

On discretizing the quality scores, what if the distribution of the scores are very skew? What issues can you see in this case and how would you resolve them?

Is it possible that the quality score won't be generated during sampling in the prediction approach because it's not guaranteed? How do you handle this situation?

---

> ### Comment · Reviewer_9pCm · 2023-11-11
> **See question 2 from reviewer 2 and weakness 2 from reviewer 3**
>
> - This weakness is straightforward and I believe this paper is not well enough to be accepted.

---

> ### Author Response · Authors · 2023-11-22
>
> We thank the reviewer for their constructive comments and answer their questions below.
>
> 1. We recommend quality prompting. Although, we did not explicitly test our method on large scale multilingual tasks, the same methods could in principle be used. In particular our experiments in the appendix in table 3 indicate that using the proposed approach for finetuning LLMs on small amounts of data works well. Since LLMs are known to be proficient in multilingual tasks, we are confident that our approach performs similarly as on the tasks mentioned in this paper. We leave it for future work to explore the applicability of the proposed approach in multilingual as well as low-resource settings.
>
> 2. This is a very important point, which we encountered during our experiments. The distribution of quality scores is in fact heavily skewed to the left. We tackled this problem by applying a equal mass binning scheme (Eq. 3 in the paper), which led to a balanced training set w.r.t. quality scores. We will include the frequency distribution plot in the camera ready version.
>
> 3. We investigated the behaviour of the model regarding possible cases where the model does not produce a quality score. However, we found that the model is able to learn to predict a quality score as early as after 500 steps. We found only a few corner cases where the model does not predict a quality score when we use epsilon sampling and sample 1000 sentences. In this case we assign a score of 0 (lowest bin) to a sample without a quality score.